# Polysaccharides from *Nitraria retusa* Fruit: Extraction, Purification, Structural Characterization, and Antioxidant Activities

**DOI:** 10.3390/molecules28031266

**Published:** 2023-01-28

**Authors:** Lijun Song, Shiqi Liu, Li Zhang, Leiqing Pan, Long Xu

**Affiliations:** 1College of Food Science and Technology, Hebei Normal University of Science & Technology, Qinhuangdao 066600, China; 2College of Food Science and Technology, Nanjing Agricultural University, Nanjing 210095, China; 3College of Food Science and Technology, Henan Agricultural University, Zhengzhou 450002, China

**Keywords:** *Nitraria retusa* fruit, polysaccharides, ultrasonic-assisted extraction, structure characterization, antioxidant activity

## Abstract

Polysaccharides are important bioactive components of *Nitraria retusa* fruit (NRF). In this study, the ultrasonic-assisted extraction (UAE) conditions of polysaccharides from *Nitraria retusa* fruit (NRFPs) were optimized by response surface methodology (RSM). The structural characteristics and antioxidant activity were investigated. The maximum NRFPs yield of 3.35% was obtained under the following optimal conditions: temperature of 59.5 °C, time of 30.5 min, liquid-to-solid ratio of 19.5 mL/g. Three polysaccharide fractions, NRFP-1 (20.01 kDa), NRFP-2 (28.96 kDa), and NRFP-3 (67.45 kDa), were isolated. Glucose, galactose, and arabinose in different percentages were identified as the primary monosaccharide units. The Fourier transform infrared spectrometer (FT-IR) and nuclear magnetic resonance (NMR) analysis indicated the presence of α- and β-glycosidic bonds in NRFPs. The NRFP-3 exhibited the highest scavenging activities against DPPH, ABTS, -OH free radicals, and Fe^+3^-reducing activity.

## 1. Introduction

The genus *Nitraria* (*Zygophllyaceae*) is widely cultivated in drought and saline–alkali soil of northwest China [1]. In some Asian countries (China, Tunisia, Egypt, Jordan, etc), *Nitraria retusa* are recognized as medicine and food homologous plants. The extracts from *Nitraria retusa* fruit (NRF) and leaves have been recognised to be beneficial for the treatment of hypertension, diabetes, irregular menstruation, neurasthenia, and dyspepsia for more than one thousand years [1,2,3]. The pharmacological activities are attributed to plentiful antioxidant compositions, including polysaccharides, polyphenols, flavonoids, alkaloids, etc. [1,2,3]. It is documented that polysaccharides from a variety of plants have significant biological and therapeutic activities, including antiproliferation, antihyperlipidemic, antitumor effects, and prevention of cardiovascular and cerebrovascular diseases [4,5,6]. The latest research showed that natural polysaccharides also contribute to their health-beneficial functions via intestinal fermentation, by way of slowing gastric emptying, working as substrates for microbial fermentation, improving bowel function, modulating the gut microbe structure, and protecting the immune system [5,6]. Specifically, the polysaccharides from NRF demonstrated significant antitumor and anti-inflammatory activities against MCF-7 cells and COX-2 cells [3,4]. 

The yield, physicochemical properties, structure, and pharmacological activities of polysaccharides are significantly affected by the extraction procedures [6,7]. Ultrasound-assisted extraction (UAE) has been recognized as an efficient and environmentally-friendly extraction method for lower energy consumption, shorter extraction time, and higher extraction yield due to the ultrasonic cavitation effect [6,7,8]. Response surface methodology (RSM) is an ideal method for multi-factors procedure optimization, because of its fewer runs of experiments and better predictability [9]. 

In this study, UAE was performed to extract polysaccharides from NRF. The extraction conditions were optimized by RSM. The structural characteristics and antioxidant activities in vitro of NRFP were also evaluated. The findings will provide a theoretical basis for further application of NRFP in the industries of functional food, cosmetics, and pharmaceuticals.

## 2. Results and Discussion

### 2.1. Model Fitting

Optimization of the UAE procedure was performed by RSM. According to the BBD, 17 runs of experiments were carried out. The results are shown in Table 1.

Design-Expert software was used to perform RSM regression analysis, and a multivariate secondary model was obtained. The constructed model expresses the extraction yield of crude NRFP (*Y*) according to the following second-order polynomial Equation (1):*Y* = −12.538 + 0.398 *X*_1_ + 0.263 *X*_2_ − 0.0004 *X*_3_ − 0.0015 *X*_1_*X*_2_ + 0.0017 *X*_1_*X*_3_ + 0.0011 *X*_2_*X*_3_ − 0.0033 *X*_1_*^2^* − 0.0033 *X*_2_*^2^* − 0.0034 *X***_3_***^2^*(1)

As shown in Table 2, the F value of 53.53 indicated that the model was significant. In this case, the terms *X*_1_, *X*_3_, *X*_1_*X*_2_, *X*_1_*X*_3_, *X*_1_^2^, *X*_2_^2^ and *X*_3_^2^ were significant. The “lack of fit (*F* = 2.96)” was not significant relative to the pure error, and the “R_Pred_^2^ (0.8350)” was in reasonable agreement with the “R_Adj_^2^ (0.9673)”. The “adeq precision ratio (23.995)”, representing the signal-to-noise ratio indicated an adequate signal. The model could be used to navigate the design space.

It can be seen that 3D response surfaces provided an intuitive presentation of the interactions between various factors and the NRFPs yield (Figure 1). The 3D pictures indicated that the NRFPs yield was significantly affected by the ultrasound temperature, time, and liquid-to-solid ratio. The results were consistent with the ANOVA results (Figure 1a–c).

### 2.2. Model Validation

Based on the feasibility of practical operation and the results of data analysis, the optimal extraction conditions were as follows: temperature of 59.5 °C, time of 30.5 min, and liquid-to-solid ratio of 19.5 mL/g. To verify the feasibility of the conditions, five more validation experiments were conducted and the average yield was 3.35 ± 0.12%, which was not significantly different from the predicted value (3.33%). The result proved the high precision and reliability of this method.

### 2.3. Thermal Characteristic of Crude NRFP

The thermal stability of polysaccharides is influenced by their structures and functional groups [10,11]. Figure 2 illustrates the thermal stability of crude NRFP. The thermal characteristic curve showed a three-step degradation pattern. The first stage was mainly attributed to a loss of moisture and dehydration (25–150 °C). Then a rapid and maximum mass loss (74.65%) appeared in the range of 150–500 °C, attributed to the changes in the polysaccharide structure. The results demonstrated that the chemical properties of crude NRFP were stable when the temperature was lower than 150 °C. Four obvious endothermic peaks were observed in the DSC curve. The first peak appeared at 115 °C, which was attributed to moisture evaporation. With the increment in temperature, three endothermic processes occurred at 307 °C, 446 °C, and 565 °C, with a maximal peak at 446 °C, which was attributed to a violent decomposition reaction, resulting in the cleavage of carbon chains and hydrogen bonds [10,11]. These observations were consistent with the results of TGA.

### 2.4. Chemical Composition of Crude NRFP

As listed in Table 3, the contents of sugar and uronic acids were 72.58% and 18.98%, respectively, which indicated that the dominant polysaccharides in NRF were acidic polysaccharides [12]. The protein content was found at a content of 8.69%. These results indicated the existence of a polysaccharide–protein complex, which showed favorable biological activities [12]. In addition, the conjugated polyphenolics, which were demonstrated to be effective antioxidants and antidiabetics, were found at a level of 6.32 mg GAE/g [13,14].

### 2.5. Structural Characterization of NRFPs

#### 2.5.1. Fractions Purification

The crude NRFPs were purified and separated by DEAE Sepharose Fast Flow and three independent elution peaks were observed (Figure 3a). The pure polysaccharides eluted with deionized water, 0.1 mol/L NaCl, and 0.3 mol/L NaCl were designated as NRFP-1, NRFP-2, and NRFP-3, respectively. The recovery rates of the three fractions were 16.23%, 15.84%, and 12.56%, respectively.

#### 2.5.2. Molecular Weight and Zeta Potential of NRFPs

It was reported that the *M_W_* of polysaccharides from plants was in the range of 10 kDa to 150 kDa [15]. The *M_W_* of polysaccharides significantly affects their physicochemical properties [11]. As illustrated in Table 4, the *M_W_* of NRFP-1, NRFP-2, and NRFP-3 were 20.01 kDa, 28.96 kDa, and 67.45 kDa, respectively. Broadband peaks and low PDI values (1.39, 1.30, and 1.76) highlighted that they were heterogeneous polysaccharides [16]. 

Zeta potential is an effective indicator of the stability of colloidal dispersions [10]. The results showed that all three fractions had negative charges. NRFP-3 exhibited larger zeta potential compared with NRFP-1 and NRFP-2, which indicated the higher stability of the NRFP-3 solution. The negative charges of polysaccharides were mainly caused by uronic acid, which was consistent with the result of the monosaccharide compositions. 

#### 2.5.3. Monosaccharide Composition of NRFPs

As shown in Figure 3b and Table 5, all three NRFPs were composed of similar monosaccharides, including Fuc, Ara, Rha, Gal, Glu, Xyl, Man, GalA, and GluA, with different percentages. Therefore, NRFPs were acidic heteropolysaccharides [17,18]. 

Glu, Gal, and Ara were the most abundant monosaccharides. The contents of Glu, Ara, and Gal in NRFP-1 were 47.22%, 21.28%, and 15.97%, respectively. Gal was the predominant monosaccharide in NRFP-2, followed by Ara and Glc. Ara was the predominant monosaccharide in NRFP-3, followed by Gal and Glc. These results were slightly different from previous studies. Rjeibi et al. [3] reported that the predominant monosaccharides were Glc (41.4%) and GalA (30.5%) in NRFP from Tunisia. Zhao et, al [18] reported that the NRFP fractions isolated from NRF in Gansu province were mainly composed of Glc. 

Therefore, NRFPs vary in monosaccharide compositions, which might be attributable to the differences in plant species and states, separation and purification methods, and polysaccharide structures [19]. Till now, studies on the extraction, structure characterization, and biological activity of *Nitraria* polysaccharides are still very limited compared with alkaloids, polyphenols, and flavonoids [20]. 

#### 2.5.4. FT-IR Spectrum Analysis of NRFPs

The functional groups of polysaccharides are usually characterized by FT-IR spectroscopy. As shown in Figure 3c, the polysaccharide fractions showed similar FT-IR spectra. 

The broad absorption peaks at 3370 cm^−1^ were attributed to the O-H stretching vibration, and the peaks at 2937 cm^−1^ were attributed to the -CH stretching vibrations, including -CH, -CH_2_, and -CH_3_ in polysaccharides [9]. The sharp band at 1630 cm^−1^ was caused by the stretching vibration of O-H [21]. The absorption peaks at 1400 cm^−1^ represented the bending vibrations of C-H and C=O in, uronic acids [22]. This result was in agreement with the monosaccharide composition. The absorption signals at 1260 cm^−1^ and 1074 cm^−1^ were attributed to C-O-H and C-O-C in pyranose sugars [23]. The spectra between 1200 cm^−1^ and 800 cm^−1^ were typical fingerprints for carbohydrates, including C-O-C glycosidic bond vibrations and C-O-H links [3]. The signals in the range of 1200–1100 cm^−1^ were caused by the stretching vibration of the glycosidic bond. The adsorption at 868.47 cm^−1^ confirmed the existence of alpha configurations [24]. Moreover, the peak at 763 cm^−1^ and 630 cm^−1^ might be attributed to the existence of *β*-configuration of pyranoses [3]. The characteristic absorptions at 892 cm^−1^ and 826 cm^−1^ indicated the possible presence of *α*-pyranose and *β*-pyranose [9,24]. It should be noted that the peak of NRFP-3 at 1740 cm^−1^ might be the characteristic absorption peak of the ester group or *O*-acetyl group [9].

#### 2.5.5. NMR Spectrum Analysis of NRFPs

As shown in Figure 4, the three purified polysaccharides displayed some differences in ^1^H and ^13^C NMR spectra. The ^1^H and ^13^C-NMR spectra of the three fractions exhibited characteristic signals of polysaccharides. The peak at δ 4.70 ppm was ascribed to D_2_O. The crowded signals of ^1^H-NMR in the range of δ 3.25–5.30 ppm implied the existence of similar monosaccharide residues [25]. The signals in the ranges of 4.4–5.4 ppm and 95.0–110.0 ppm were assigned to the typical characteristic signals of polysaccharides [26]. 

For NRFP-1(Figure 4a,b), the chemical shifts of anomeric hydrogen at δ 5.38, 5.16, 5.03, 4.58, 4.56 ppm and the anomeric carbon signals at δ 98.09, 95.93, 95.37, 92.12 ppm revealed that NRFP-1 might mainly contain five kinds of monosaccharide residues and confirmed the presence of both *α*- and *β*-configurations. The result was in agreement with the presence of FT-IR bands at 843 cm^−1^ [27]. According to previous studies [27,28,29], the ^1^H signal at δ 5.03 ppm and ^13^C signal at 69.23 and 69.72 ppm indicated the presence of *α*-(1→6) chain-expending anomeric signal. The ^1^H signal at δ 5.16 and 4.56 ppm corresponded to *α*-Araf and 1,3-Galp, respectively. The peaks at 74.19, 75.98, 69.72, and 60.67 ppm for NRFP-1 were attributed to *β*-D-glucose. The anomeric proton chemical shift of NRFP-2 at δ 5.17 ppm confirmed the presence of H-1 of *α*-Araf (Figure 4c,d). The ^1^H signals at δ 1.25 were assigned to the methyl group in rhamnosyl residues. The anomeric carbon chemical shift at δ 100.35 ppm represented C-1 of 1,2-*α*-L Rha. The peak at δ 62.48 was attributed to C-6 of 1,3-*β*-D-Gal [27,28,29]. The presence of uronic acid in NRFP-2 was confirmed according to the chemical shifts at δ 179.78 and δ 181.06. The anomeric proton chemical shift of NRFP-3 at δ 5.02 ppm indicated the presence of the *α*-(1→6) chain. The anomeric proton signal at δ 5.07 ppm and carbon signal at δ 107.53 ppm were attributed to the presence of H-1 and C-1 of *α*-1,5-Araf (Figure 4e,f). The anomeric peaks at δ 109.2 ppm and δ 76.73 ppm corresponded to *α*-1,5-Ara and *β*-D-Glc, respectively. The signal at δ 175.45 ppm confirmed the presence of C-6 of -1,4-GalpA [27,30,31]. 

#### 2.5.6. SEM Analysis of NRFPs

As presented in Figure 5, different NRFP fractions were easily distinguishable according to their microscopic characteristics. Crude NRFPs showed a relatively uniform sphere with aggregation, which was similar to those of some okra polysaccharide fractions [32]. The surfaces of NRFP-1 and NRFP-2 showed sheet structures, with irregular bulges and some ruptures. Moreover, NRFP-2 showed a rough surface compared with NRFP-1, which was in agreement with the findings reported by Abuduwaili et al. [4]. NRFP-3 was found to be in an irregular shape with a rough surface and aggregated sphere. These inconsistent findings might be attributable to different extraction and purification procedures [33].

### 2.6. Antioxidant Activity of NRFPs

The overproduction of free radicals and oxidative stress are important factors causing aging, cancer, diabetes, and cardiovascular diseases [34]. The potential of NRFPs for radical scavenging was investigated in the present study. The results are shown in Figure 6. The crude NRFPs and purified fractions all showed scavenging capabilities against DPPH, ABTS, and -OH radicals in a dose-dependent manner. Notably, the crude NRFPs showed stronger scavenging capability compared with each purified fraction. Among the three fractions, NRFP-3 showed the highest scavenging activities against DPPH, ABTS, and -OH radicals, with scavenging rates of 37.13%, 75.50%, and 45.90% at 1.0 mg/mL, respectively. The reducing capacities of different polysaccharides are shown in Figure 6d. Similarly, crude NRFPs and NRFP-3 also exhibited higher reducing activities compared with other fractions. It is worth noting that the antioxidant activities of all polysaccharide components are significantly lower than that of Vc.

The physiological activities of polysaccharides are closely related to their structural properties [11]. However, few studies are available about the structure–function relationship of NRFPs. Therefore, a preliminary inference about the relationship could be drawn based on the previously published studies. As reported, the antioxidant activities of polysaccharides were positively correlated with the molar ratio of Ara, Rib, Man, and Glc. In contrast, the correlation between antioxidant activity and Fru and Gal was negative [35]. Moreover, it was discovered that D-mannose could affect regulatory T cells and improve the detection and phagocytosis of tumor cells by the immune system [30]. The *M_w_* and water solubility of polysaccharides may have an impact on the anticancer effect. Usually, polysaccharides with higher *M_w_* and better water solubility showed stronger anticancer activity [36]. In addition, there was a higher probability that polysaccharides containing *β*-(1→4), *α*-(1→4), or *α*-(1→6) glycosidic linkages would exhibit antioxidant activities [37]. The presence and molar ratios of Glc and Man monomers may contribute to the immunomodulatory effect of polysaccharides. Possibly, polysaccharides with *β*-(1→4) glycosidic linkages or major chains with *β*-*D*-(1→3)-Glc repeats would exert immunomodulatory effects [36,38]. The polysaccharides with β-conformation and higher *M_w_* exerted stronger antioxidant activities than those with α-conformation and lower *M_w_* [39]. Additionally, for marine algae polysaccharides (MAPs), the low-molecular fractions (3.2 and 7.4 kDa) showed more effective endothelial protection than the medium-molecular-weight fractions (28 and 34 kDa), while high-molecular-weight MAPs significantly reduced the fasting blood glucose, total cholesterol, and total body fat. The MAPs with highly sulfated (29.57% and 34.02%) exhibited a better upregulating effect on the fibroblast growth factor/FGF receptor signaling in BaF3 cells [40]. 

In this paper, the strongest radical scavenging activities of NRFP-3 might depend on the highest percentage of Ara and Man, together with the highest *M_W_* [35,36]. In conclusion, the structure–activity relationships of polysaccharides and their physiological activity mechanisms still need further studies.

## 3. Materials and Methods

### 3.1. Materials and Reagents

NRF was collected from Baicheng Country, Xinjiang, China (38°42′ N, 80°96′ E) in June 2021. The fruits were cleaned, deseeded, and lyophilized. Then the lyophilized samples were ground into powder and sieved through a 60-mesh sieve, and stored at −18 °C for further analysis.

The standards (mannose, riboose, rhamnose, glucuronic acid, galacturonic acid, N-acetyl-amino glucose, glucose, N-acetyl-aminogalactose, galactose, xylose, arabinose, and fucos), deuterium oxide (D_2_O), 1-phenyl-3-methyl-5-pyrazolone (PMP, M70800-100G), anion-exchange DEAE (diethylaminoethyl) sepharose Fast Flow and dialysis bag (8000–14,000 Da) were obtained from Solarbio Science & Technology Co., Ltd. (Beijing, China). A series of dextran standards (5 k, 50 k, 150 k, 650 k, and 1100 k) were obtained from Aladdin Reagent Int. (Shanghai, China). The 1,1-Diphenyl-2-picrylhydrazyl free radical (DPPH), and 2,2′-azinobis (3-ethylbenzothiazoline-6-sulfonic acid) free radical (ABTS) were obtained from Jiancheng Technology Co., Ltd. (Nanjing, China). All other chemicals of analytical grade were purchased from Jiancheng Technology Co., Ltd. (Nanjing, China).

### 3.2. Extraction Procedure

#### 3.2.1. UAE Extraction

NRF powder (2.0 g) was mixed with distilled water (10–30 mL/g) at different temperatures (40–80 °C) for 10–60 min in an ultrasonic extractor (Scientz Biotechnology Inc., Ningbo, China). After extraction, the supernatant was concentrated and precipitated with four times the volume of ethanol to obtain the precipitate. The precipitate was dissolved and deproteinized using the Sevage method and decolorized with H_2_O_2_ solution (30%, *v*/*v*). Then, the solution was concentrated and dialyzed (MWCO: 3000 Da) against water for 48 h. Finally, the crude NRFP was lyophilized and stored at −18 °C. The extraction yield (Y, %) was calculated as the dry weight of the crude NRFP (*m*_1_) relative to the weight of NRF powder (*m*_2_) (2):(2)$$Y(\%)=\frac{m_{1}}{m_{2}}\times 100\%$$

#### 3.2.2. Process Optimization

Box–Behnken design (BBD) was used to optimize the extraction processes for maximum extraction yield. The experimental designs were listed in Table 1. The model fitting and analysis of variance (ANOVA) were conducted using Design-Expert 8.0.6 (Stat-Ease Inc., Minneapolis, MN, USA) according to the following equation:(3)$$Y=\delta_{0}+\sum_{m=1}^{3}\delta_{m}X_{m}+\sum_{m=1}^{3}\delta_{m}X_{mm}^{2}+\sum_{m+n}^{3}\delta_{mn}X_{mn}$$
where *δ*_0_ represents constant, *δ_i_*, *δ_ii_*, and *δ_ij_* linear, are quadratic and interactive coefficients, respectively. *X_m_*, *X_mm_*, and *X_mn_* represent the variable levels. The appropriateness was determined using coefficient of determination (*R*^2^). The significance and variables were evaluated according to *p* and *F* values [9].

### 3.3. Chemical Composition Analysis

The content of total sugar, uronic acid, protein, and total phenolic was determined by the phenol-sulfuric acid method [41], the m-hydroxy diphenyl method [41], the Bradford method [41], and the Folin–Ciocalteu method [42].

### 3.4. Isolation and Purification

A total of 10 mL of crude NRFP (20 mg/mL) was added to a DEAE sepharose Fast Flow chromatography column (1.6 cm × 20 cm) equilibrated with distilled water. Then, the crude NRFP was eluted with distilled water and various concentrations of NaCl solution (0.1, 0.2, 0.3, and 0.5 mol/L) for 200 min at a flow rate of 1.0 mL/min. The fractions were collected every 15 min, and the total sugar contents were determined by the anthracnose-sulfuric acid method. According to the eluting curve, the polysaccharide fractions were collected, concentrated, and lyophilized to yield purified fractions designated NRFP-1, NRFP-2, and NRFP-3 [43].

### 3.5. Thermal Analysis

Thermogravimetric (TG) and differential scanning calorimetry (DSC) of purified NRFPs (2.0 mg) were conducted on an STA-449-F3 thermogravimetric analyzer (Netzsch, Bavaria, Germany) under a protective nitrogen atmosphere (50 mL/min). The heating was carried out ranging from 25 °C to 700 °C with a gradient of 10 °C/min.

### 3.6. Structural Analysis

#### 3.6.1. Monosaccharide Composition

The high-performance liquid chromatography (HPLC) with a 1-phenyl-3-methyl-5-pyrazolone (PMP) derivatization method was used to analyze the monosaccharide composition [44]. 5 mg of NRFP was hydrolyzed with 5 mL of trifluoroacetic acid (TFA, 3.0 mol/L) at 110 °C for 8 h. Then, the NRFP was dried under nitrogen, and dissolved in 300 μL of distilled water. A total of 250 μL of NRFP hydrolysates (1.0 mg/mL) was mixed with 250 μL of NaOH solution (0.6 mol/L) and 500 μL of PMP-methanol solution (0.4 mol/L). Then, the mixed solutions were reacted at 70 °C for 60 min in darkness. After being cooled in cold water for 10 min, 100 μL of HCl solution (0.3 mol/L) was added to terminate the reaction. Then, 1.0 mL of chloroform was added and vortexed vigorously, centrifuged at 10,000 r/min for 8 min. The supernatant was analyzed by HPLC.

The analysis was performed using the Ultimate 3000 HPLC system equipped with a diode array detector and a C18 column (4.6 mm × 200 mm, 5 μm). The column temperature was 30 °C, and the detection wavelength was 250 nm. The mobile phases were 83% of phosphate buffer (pH 6.7, 50 mol/L) and 17% of acetonitrile at a flow rate of 1.0 mL/min. The monosaccharide standards (2.0 mg/mL) were used for qualitative identification. 

#### 3.6.2. Molecular Weight Analysis

The molecular weight (*M_W_*) was analyzed by high-performance gel permeation chromatography (HPGPC) using a Waters 1515 system equipped with a refractive index detector (1260 RID, Agilent Technologies, Santa Clara, CA, USA), and a G4000 SWXL column (4.6 mm × 30 cm, Tosoh Biosep, Montgomeryville, PA, USA). The HPGPC analysis was performed at 30 °C. 10 μL of NRFP (2 mg/mL) was eluted with KH_2_PO_4_ solution (0.1 mol/L) at a flow rate of 1.0 mL/min. Calibration was performed relative to dextran standards via a calibration curve [45]. 

#### 3.6.3. Fourier Transform Infrared Spectrometer (FT-IR) Analysis

The NRFP (1.0 mg) was mixed with KBr (150 mg), ground, and compressed into flakes. FT-IR spectrum was recorded using a Nicolet 5700 FT-IR Spectrometer (Thermo Electron, Madison, WI, USA) at room temperature with a range of 4000–400 cm^−1^ [45]. 

#### 3.6.4. Nuclear Magnetic Resonance (NMR) Analysis

The NRFP (10 mg) was suspended in 500 mL of D_2_O (99.96%) and freeze-dried twice before being dissolved in 500 mL of high-quality D_2_O. The ^1^H and ^13^C NMR spectra were collected by an ADVANCE III-600 spectrometer (Burker, Karlsruhe, Germany) at room temperature [46].

#### 3.6.5. Determination of Zeta-Potential

The NRFP samples were dispersed in phosphate buffer (50 mmol/L, pH 7.0) to form a polysaccharide solution (0.2%, *w*/*w*). The zeta potential of NRFP solution (1.0 mL) was measured at 25 °C using a Zeta-PALS instrument (Zetapals, Brookhaven Instruments, New York, NY, USA).

#### 3.6.6. Scanning Electron Microscopy (SEM) Analysis

The NRFP powder was fixed onto a copper stub. After sputtering with a layer of gold, the S-4800 SEM (JEOL Ltd., Tokyo, Japan) was used to record the SEM images at an accelerating voltage of 20 kV [45].

### 3.7. Antioxidant Activities Analysis

The DPPH and ABTS radical scavenging activities of NRFPs were measured according to the reported method [47]. The hydroxyl radical scavenging abilities and the Fe^+3^ reducing power were determined according to the method described previously [10]. 

### 3.8. Statistical Analysis

All data are shown as the mean ± standard deviations from three replicates. Statistical significance was conducted using SPSS 22.0 (*p* < 0.05 represents statistical significance).

## 4. Conclusions

In this study, NRFPs were extracted by UAE effectively, with three purified polysaccharide fractions (NRFP-1, NRFP-2, and NRFP-3) being obtained. The NRFPs were composed of Man, Rib, Rha, GluA, GalA, Glc, Gal, Xyl, Ara, and Fuc with different percentages. The structural analysis by FT-IR and NMR highlighted the α- and β-glycosidic linkages in NRFPs. NRFPs showed irregular shapes with rough surfaces and aggregated spheres. In addition, NRFPs exhibited strong radical scavenging activities and Fe^3+^-reducing activity. These results will provide a theoretical foundation for the application of NRFPs in functional foods and medicines.

## Figures and Tables

**Figure 1 molecules-28-01266-f001:**
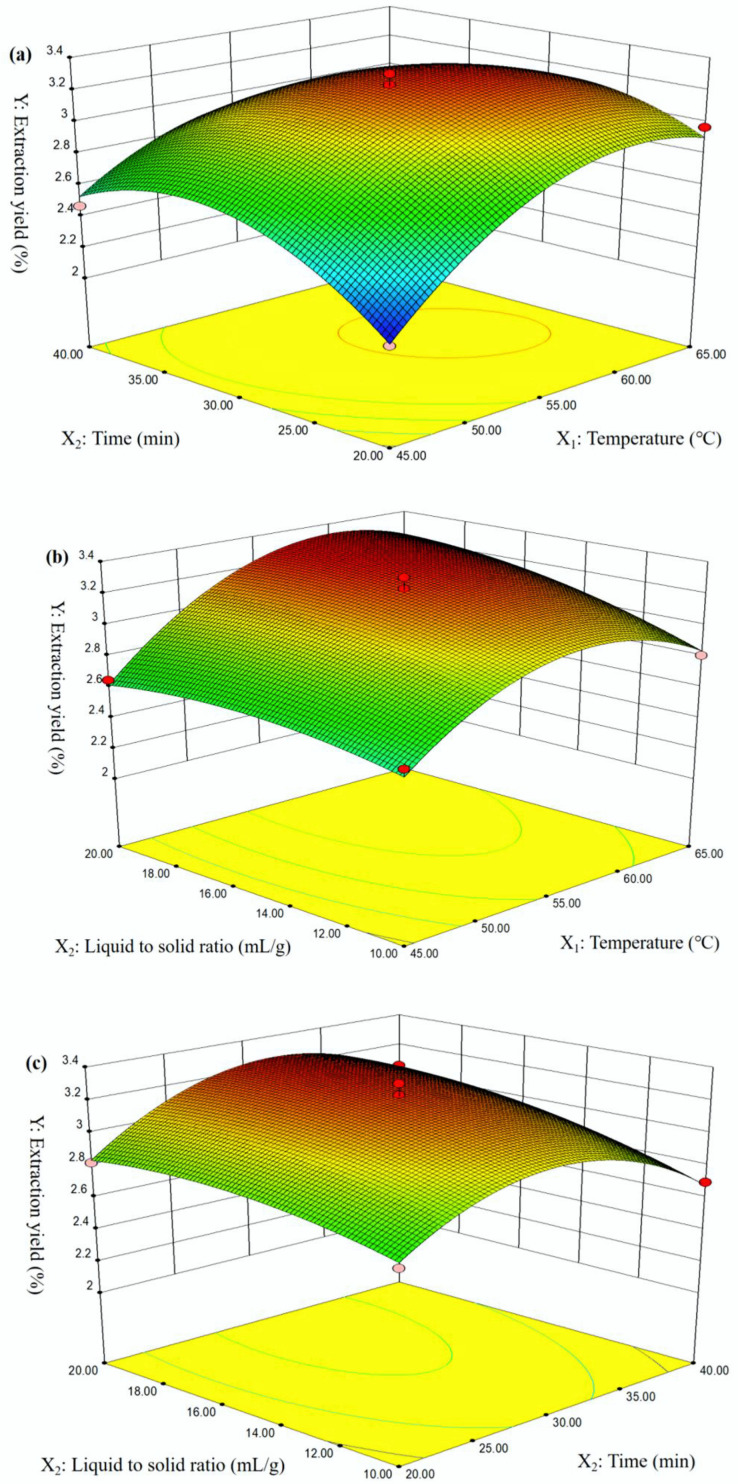
Effect of temperature (*X*_1_), time (*X*_2_), and liquid-to-solid ratio (*X*_3_) on the NRFP yield.

**Figure 2 molecules-28-01266-f002:**
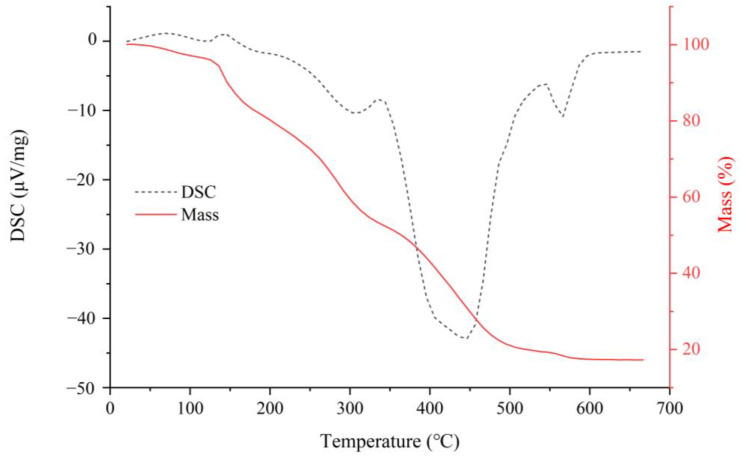
The TGA thermogram of crude NRFP.

**Figure 3 molecules-28-01266-f003:**
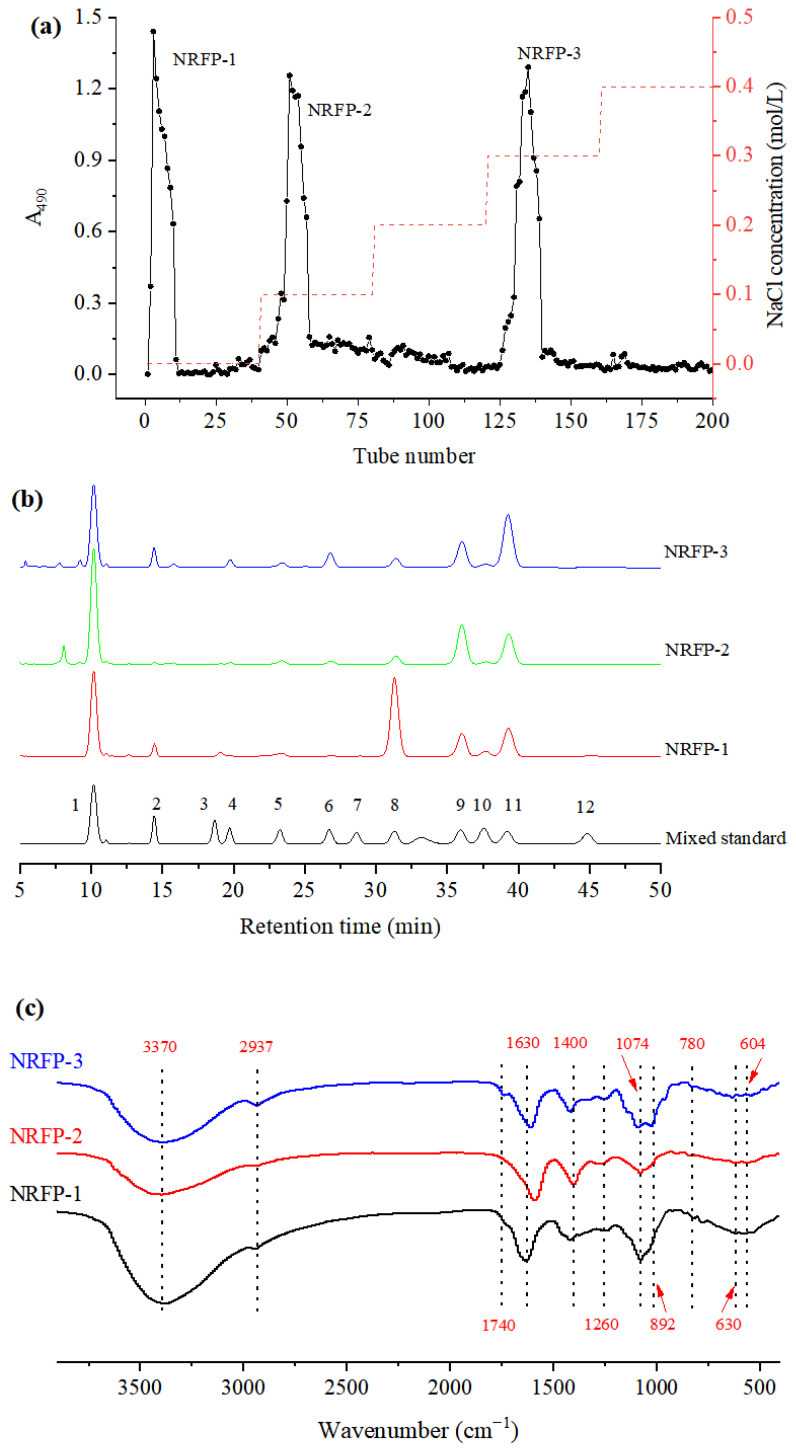
The elution curve (**a**), HPLC chromatograms (**b**), and FT-IR spectrum of NRFPs (**c**). 1: Mannose (Man); 2: Ribose (Rib); 3: Rhamnose (Rha); 4: Glucuronic acid (GluA); 5: Galactose acid (GalA); 6: N-acetyl-amino glucose (N-Glu); 7: Glucose (Glu); 8: N-acetyl-aminogalactose (N-Gal); 9: Galactose (Gal); 10: Xylose (Xyl); 11: Arabinose (Ara); 12: Fucose (Fuc).

**Figure 4 molecules-28-01266-f004:**
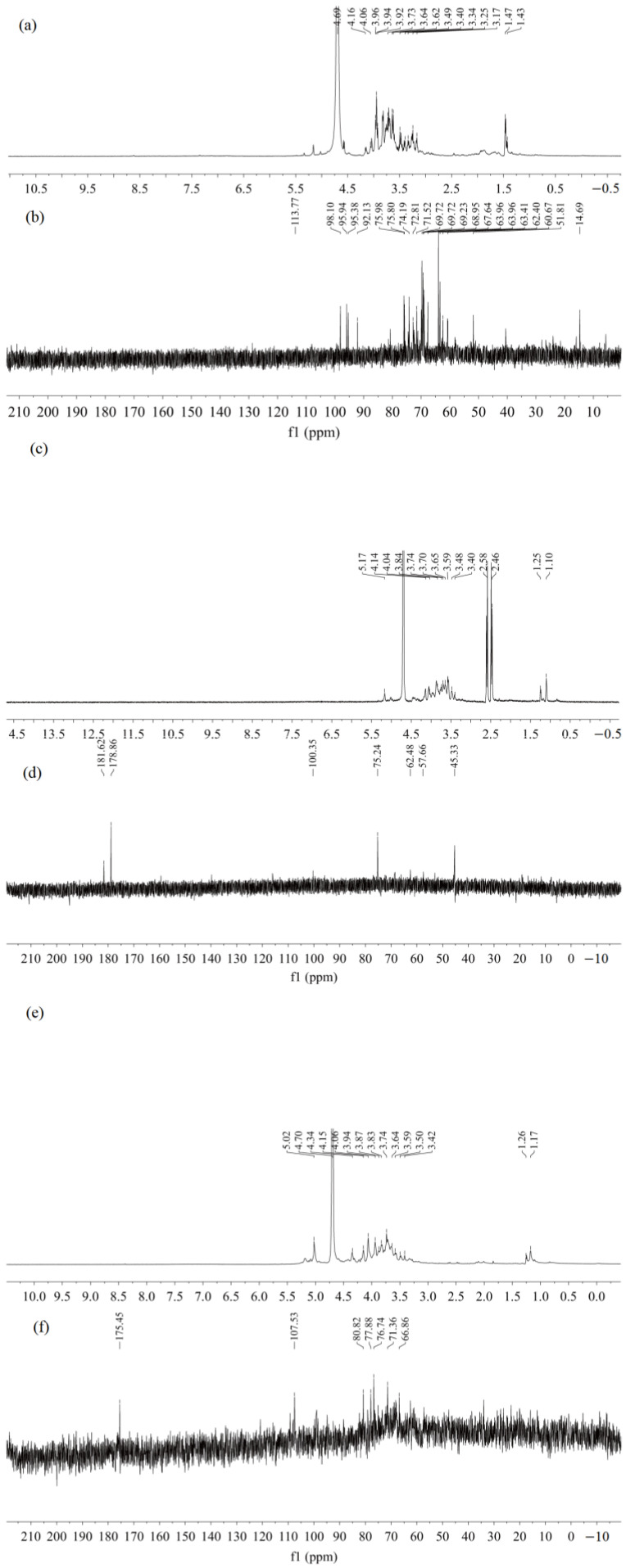
^1^H and ^13^C NMR spectra of NRFPs. (**a**) ^1^H NMR spectra of NRFP-1; (**b**) ^13^C NMR spectra of NRFP-1; (**c**) ^1^H NMR spectra of NRFP-2; (**d**) ^13^C NMR spectra of NRFP-2; (**e**) ^1^H NMR spectra of NRFP-3; (**f**) ^13^C NMR spectra of NRFP-3.

**Figure 5 molecules-28-01266-f005:**
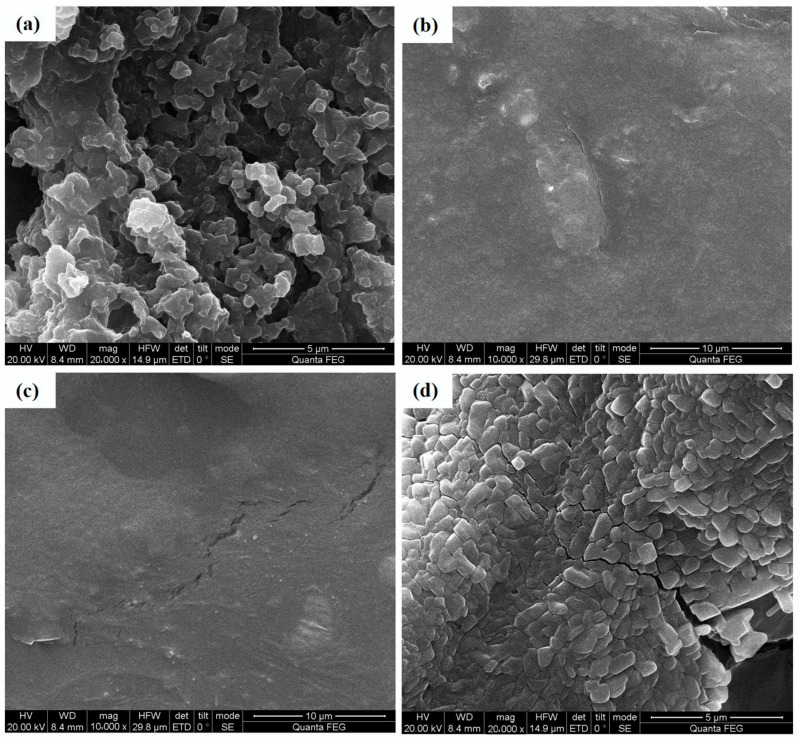
The scanning electron microscopy of NRFPs. (**a**) CNRFP; (**b**) NRFP-1; (**c**) NRFP-2; (**d**) NRFP-3.

**Figure 6 molecules-28-01266-f006:**
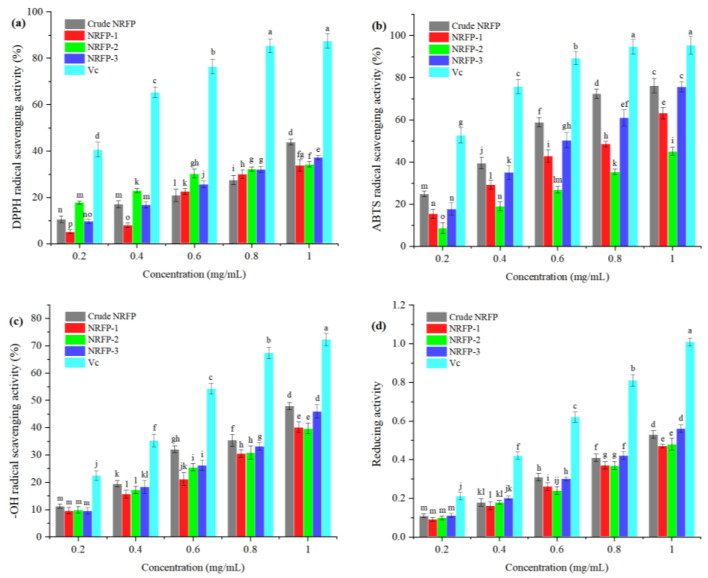
The antioxidant activities of NRFPs. (**a**) DPPH radical scavenging activity; (**b**) ABTS radical scavenging activity; (**c**) -OH radical scavenging activity; (**d**) reducing activity.

**Table 1 molecules-28-01266-t001:** RSM design and results.

No.	*X*_1_: Temperature (°C)	*X*_2_: Time (min)	*X*_3_: Liquid-to-Solid Ratio (mL/g)	*Y*: Extraction Yield (%)
1	65	30	20	3.20
2	55	40	10	2.70
3	55	30	15	3.30
4	55	20	20	2.82
5	45	30	10	2.60
6	55	30	15	3.20
7	65	20	15	2.97
8	45	30	20	2.65
9	65	40	15	2.71
10	55	30	15	3.23
11	55	30	15	3.20
12	45	20	15	2.15
13	45	40	15	2.47
14	65	30	10	2.81
15	55	40	20	3.06
16	55	30	15	3.20
17	55	20	10	2.68

**Table 2 molecules-28-01266-t002:** ANOVA for the quadratic model.

Source	Sum of Squares	df	Mean Square	*F*	*p*
Model	1.67	9	0.19	53.53	<0.0001
*X* _1_	0.41	1	0.41	119.67	<0.0001
*X* _2_	0.013	1	0.013	3.7	0.0959
*X* _3_	0.11	1	0.11	31.92	0.0008
*X* _1_ *X* _2_	0.084	1	0.084	24.31	0.0017
*X* _1_ *X* _3_	0.029	1	0.029	8.35	0.0233
*X* _2_ *X* _3_	0.012	1	0.012	3.5	0.1037
*X* _1_ ^2^	0.45	1	0.45	128.93	<0.0001
*X* _2_ ^2^	0.45	1	0.45	128.93	<0.0001
*X* _3_ ^2^	0.031	1	0.031	8.9	0.0204
Residual	0.024	7	3.46 × 10^−3^		
Lack of fit	0.017	3	5.57 × 10^−3^	2.96	0.1608
Pure error	7.52 × 10^−3^	4	1.88 × 10^−3^		
Cor. total	1.69	16			
Adeq precision	1.67	9	0.19	53.53	<0.0001
C.V%	0.41	1	0.41	119.67	<0.0001
R^2^ = 0.9719, Adj-R^2^ = 0.9673, Pred-R^2^ = 0.8350					

**Table 3 molecules-28-01266-t003:** Chemical compositions of crude NRFPs.

Total Sugar Content (%)	Uronic Acid Content (%)	Protein Content (%)	Total Polyphenol Content (mg GAE/g)
72.58 ± 2.68	18.98 ± 1.50	8.69 ± 1.66	6.32 ± 0.86

**Table 4 molecules-28-01266-t004:** Molecular weight distribution of NRFPs.

Samples	*M_W_* (kDa)	*Mn* (kDa)	*Mp* (kDa)	PDI (*M_W_*/*Mn*)	*Zeta* Potential
NRFP-1	20.01	8.75	12.15	1.39	−9.27
NRFP-2	28.96	12.46	16.24	1.30	−10.9
NRFP-3	67.45	48.97	55.69	1.76	−19.8

Abbreviations: *M_w_*: weight average molecular weight, *Mn*: number average molecular weight, *Mp*: peak average molecular weight, PDI: polydispersity index.

**Table 5 molecules-28-01266-t005:** Monosaccharide compositions of NRFPs (%).

NRFPs	Man	Rib	Rha	GluA	GalA	Glc	Gal	Xyl	Ara	Fuc
NRFP-1	3.73 ± 0.21	2.01 ± 0.22	0.46 ± 0.10	2.97 ± 0.26	0.54 ± 0.07	47.22 ± 0.86	15.97 ± 0.65	4.06 ± 0.19	21.28 ± 0.72	1.51 ± 0.10
NRFP-2	0.90 ± 0.13	0.47 ± 0.11	1.24 ± 0.11	3.93 ± 0.23	3.16 ± 0.16	8.05 ± 0.37	43.07 ± 1.02	3.03 ± 0.21	36.05 ± 0.77	0.10 ± 0.03
NRFP-3	6.96 ± 0.42	ND	3.41 ± 0.23	3.40 ± 0.28	9.26 ± 0.41	6.10 ± 0.39	21.30 ± 0.79	2.95 ± 0.18	46.33 ± 0.98	0.30 ± 0.04

Abbreviations: Man, mannose; Rib, Ribose; Rha, rhamnose; GluA, glucuronic acid; GalA, galactose acid; Glc, glucose; Gal, galactose; Xyl, xylose; Ara, arabinose; Fuc, fucose.

## Data Availability

Data are contained within the article.

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
