# Peer review of "Polysaccharides from *Nitraria retusa* Fruit: Extraction, Purification, Structural Characterization, and Antioxidant Activities"

_molecules, 2023, doi:10.3390/molecules28031266_

Round 1

Reviewer 1 Report

1- page 1 line 17:  The Maximum yield of what?

2- page 1 line 20 : where the concentration % with Standard diviation of Glucose .......

3- page 1 line 23: what is the meaning of  DPPH, ABTS, -OH

4-page2 lines 51-52:  The findings will provide a theoretical basis for further application of NRFP in the industries of functional food, cosmetics, and pharmaceuticals.  these sentence  should be omit from here and can be add in the conclusion.

5- page 6 line 111-112: in figure 3 what is the meaning of Man;  Rib;  Rha; GlcA, GalA; N-Glc; Glc; N-Gal; Gal;  Xyl; Ara; Fuc.

6- Page 11, Figure 6: In the figure 6  which explain the antioxidant activities for NRFP where the positive control such as (BHA or BHT). it is very important to add the positive controle to make agood comparison .

7- page 14 lines 334 -344: the conclusion need to be  improved because it is not containe any numerical results which it presented  in the results and discussion 

Author Response

Dear editor

Thank you very much for your kind comments and constructive revision suggestions on our manuscript “Polysaccharides from Nitraria retusa fruit: extraction, purification, structural characterization, and antioxidant activities” (molecules-2112440). As you and the reviewers suggested, we have edited our manuscript carefully with the help of a language editing company. We have also revised the manuscript according to the reviewers’ comments. We hope the revisions we made (marked in RED in the resubmitted manuscript) can answer you well. The detailed responses to the reviewers are also uploaded. If you have any queries, please don’t hesitate to contact us.

Best regards,

Lijun Song (slj176@163.com).

December 28, 2022.

Comments and Suggestions for Authors

Reviewer 1

Point 1: Page 1 line 17:  The Maximum yield of what?

Response 1: Thanks for your good question. In the revised manuscript, we illustrated the maximum yield of Nitraria retusa fruit polysaccharides (NRFPs).

Point 2: Page 1 line 20: where the concentration % with Standard diviation of Glucose .......

Response 2: Thanks for your good question. The concentration of monosaccharides was listed in Table 5.

Point 3: Page 1 line 23: what is the meaning of DPPH, ABTS, -OH.

Response 3: The DPPH, ABTS, and -OH are abbreviations of three free radicals, including Hydroxyl free radical (-OH), 1,1-Diphenyl-2-picrylhydrazyl free radical (DPPH), and 2,2’-azino bis(3-ethylbenzothiazoline-6-sulfonic acid) free radical (ABTS). In this paper, we used free radical scavenging activities to evaluate the antioxidant activities of NRFPs. We have added the details of these free radicals in Lines 252-254 in the revised manuscript. Thank you very much.

Point 4: Page 2 lines 51-52: The findings will provide a theoretical basis for further application of NRFP in the industries of functional food, cosmetics, and pharmaceuticals. These sentences should be omitted from here and can be added to the conclusion.

Response 4: Thanks for your suggestion. We have deleted that sentence in the Abstract section and added in the Conclusion section (in Lines 356-357).

Point 5: Page 6 lines 111-112: in figure 3 what is the meaning of Man;  Rib; Rha; GlcA, GalA; N-Glc; Glc; N-Gal; Gal; Xyl; Ara; Fuc.

Response 5: Thanks for your suggestion. We have added the details of these abbreviations in Figure 3 (in Lines 112-113, and 245-247).

Point 6: Page 11, Figure 6: In figure 6, which explains the antioxidant activities for NRFP where the positive control such as (BHA or BHT). It is very important to add positive control to make a good comparison.

Response 6: Thanks for your good suggestion. Yes, we agree with you, and this is a significant point of view. In the revised manuscript, we added ascorbic acid (Vc) as a positive control (in Figure 6) and revised the Discussion section (in Lines 222-223). Thank you very much.

Point 7: Page 14 lines 334-344: The conclusion need to be improved because it is not contained any numerical results which it presented in the results and discussion. 

Response 7: Thanks for your good suggestion. We revised the Conclusion section carefully according to your constructive suggestions (in Lines 349-356).

Reviewer 2 Report

In this manuscript of "Polysaccharides from Nitraria retusa fruit: extraction, purification, structural characterization, and antioxidant activities", it used UAE to extract polysaccharides and evaluated its structural characteristics and antioxidant activities. It is certainly contributing to the current scientific knowledgebase. But this needs revision and the author reply comments properly before accepted. The comments and questions are as follows:

1. What is the actual scientific problem that the MS really try to solve? Is there a plan or practice to put this conclusion further into industrial scale application?

2. Are there plans to consider testing for other biological activities besides antioxidant activity? Why was the antioxidant activity prioritised and did the authors forget to mark the significance analysis in Fig. 6?

3. The reference to biological activities of polysaccharide in the MS could be made to more relevant literature, e.g. 10.1016/j.ijbiomac.2021.05.181, 10.3390/foods11223550, and 10.3390/nu10081055, to give credibility to the MS.

4. Why is it necessary to separate the three fractions of NRFP for treatment here? The SEM images show that the morphology of the three fractions is very different after treatment. Was there a final selection of the most optimal choice?

5. In the light of further references, please briefly describe the differences in the physicochemical properties of the three fractions that lead to differences in biochemical activities other than antioxidant activity, or the possibility of such differences.

Above all, I suggest this can be accepted before making revision or answer the questions properly.

Author Response

Dear Editor

Thank you very much for your kind comments and constructive revision suggestions on our manuscript “Polysaccharides from Nitraria retusa fruit: extraction, purification, structural characterization, and antioxidant activities” (molecules-2112440). As you and the reviewers suggested, we have edited our manuscript carefully with the help of a language editing company. We have also revised the manuscript according to the reviewers’ comments. We hope the revisions we made (marked in RED in the resubmitted manuscript) can answer you well. The detailed responses to the reviewers are also uploaded. If you and the reviewers have any queries, please don’t hesitate to contact us.

Best regards,

Lijun Song (slj176@163.com).

December 28, 2022. 

Comments and Suggestions for Authors

Reviewer 2

Point 1: What is the actual scientific problem that the MS tries to solve? Is there a plan or practice to put this conclusion further into industrial-scale application?

Response 1: Thanks for your good question. Nitraria retusa fruit (NRF) are recognized as medicine and food homologous plants in China. Polysaccharides are important bioactive components, which are related to a variety of biological activities of NRF. What is more important, the bioactive and physicochemical properties of polysaccharides are closely related to their structural characteristics. To our knowledge, There are few reports on the physicochemical and structural properties of purified polysaccharides from NRF. So, we tried to clarify the optimal extraction conditions, together with the structural properties and antioxidant activities of NRFPs in this paper. And the results might provide a theoretical foundation for the application of NRFPs in different fields.

Additionally, more research is needed to be carried out to meet the requirements of polysaccharides production on an industrial scale. And the economy, environmental protection, automation, and comprehensive efficiency still should be taken into consideration.

Point 2: Are there plans to consider testing for other biological activities besides antioxidant activity? Why was the antioxidant activity prioritized and did the authors forget to mark the significance analysis in Fig. 6?

Response 2: Thanks for your kind suggestion. Yes, we agree with you that this may be an interesting point of view. Previous studies suggested that plant polysaccharides have a variety of biological activities, such as antiproliferative activity, digestive enzyme inhibitory activity (α-Amylase and α-Glucosidase), and anti-inflammatory activity.

However, it appeared that we had merely scratched the surface thus far about the research on NRFPs. Your suggestion will be the main direction of our further research. Further studies are urgently needed on the detailed structural characteristics, in vivo physiological activities, and mechanism of NRFPs. Thank you very much.

Point 3: The reference to biological activities of polysaccharides in the MS could be made to more relevant literature, e.g. 10.1016/j.ijbiomac.2021.05.181, 10.3390/foods11223550, and 10.3390/nu10081055, to give credibility to the MS.

Response 3: Thanks for your kind suggestion. In the revised paper, we have replenished these references (References 5, 6, and 41). In addition, we also replenished some sentences according to your suggestions (in Lines 40-44, and 240-246).

Point 4: Why is it necessary to separate the three fractions of NRFP for treatment here? The SEM images show that the morphology of the three fractions is very different after treatment. Was there a final selection of the most optimal choice?

Response 4: Thanks for your good question.

Generally, the crude polysaccharide extracted from plant materials are mixed components, and the composition of polysaccharides obtained by different extraction methods show significant differences. So it is necessary to study the physicochemical properties of different purified fractions. In this manuscript, the three polysaccharide fractions were purified and separated by DEAE Sepharose Fast Flow from the crude NRFP (Figure 3a). The eluting solvents are deionized water, NaCl (0.1 mol/L), and NaCl (0.3 mol/L), respectively.

The SEM images can clearly show the surface morphology and microscopic characteristics of polysaccharides. Previous studies indicated that the microscopic characteristics of polysaccharides are closely related to their physical characteristics (Li et al., 2020), and different treatments can destroy the original connections of NRFPs, thereby causing microstructure changes and differences in physical properties (Xu et al., 2018; Zhao et al., 2017). In this manuscript, we found the morphology of the three fractions was very different, and these findings were consistent with some previous studies (Abuduwaili et al., 2021; Xiong et al., 2022). The differences in the microstructure might confer different physical properties to NRFPs and thus determine their applications in different industrial fields (Zhang et al., 2016).

References

Abuduwaili, A.; Mutailifu, P.; Nuerxiati, R.; Gao, Y.H.; Ais, H.A.; Yili A. Structure and biological activity of polysaccharides from Nitraria sibirica pall fruit. Food Biosci. 2021, 40, 100903. doi.org/10.1016/j.fbio.2021.100903.

Li, J., Niu, D.B., Zhang, Y., Zeng, X.A. Physicochemical properties, antioxidant and antiproliferative activities of polysaccharides from Morinda citrifolia L. (Noni) based on different extraction methods. Int. J. Biol. Macromol. 2020, 150, 114–121. https://doi.org/10.1016/j.ijbiomac.2019.12.157.

Xiong, G.Y.; Ma, L.S.; Zhang, H.; Li, Y.P.; Zou, W.S.; Wang, X.F.; Xu, Q.S.; Xiong, J.T.; Hu, Y.P.; Wang, X.Y. Physicochemical properties, antioxidant activities and α-glucosidase inhibitory effects of polysaccharides from Evodiae fructus extracted by different solvents. Int. J. Biol. Macromol. 2022, 194, 484–498. https://doi.org/10.1016/j.ijbiomac.2021.11.092.

Xu, Y.Q., Niu, X.J., Liu, N.Y., Gao, Y.K., Wang, L.B., Xu, G., Li, X.G., Yang, Y. Characterization, antioxidant and hypoglycemic activities of degraded polysaccharides from blackcurrant (Ribes nigrum L.) fruits. Food Chem. 2018, 243, 26–35. https://doi.org/10.1016/j.foodchem.2017.09.107.

Zhang, W.J., Huang, J., Wang, W., Li, Q., Chen, Y., Feng, W.W., Zheng, D.H., Zhao, T., Mao, G.H., Yang, L.Q., Wu, X.Y. Extraction, purification, characterization, and antioxidant activities of polysaccharides from Cistanche tubulosa. Int. J. Biol. Macromol. 2016, 93, 448–458. https://doi.org/10.1016/j.ijbiomac.2016.08.079.

Zhao, C.C., Li, X., Miao, J., Jing, S.S., Li, X.J., Huang, L.Q., Gao, W.Y. The effect of different extraction techniques on property and bioactivity of polysaccharides from Dioscorea Hemsley, Int. J. Biol. Macromol. 2017, 102, 847–856. https://doi.org10.1016/ j.ijbiomac. 2017.04.031.

Point 5: In the light of further references, please briefly describe the differences in the physicochemical properties of the three fractions that lead to differences in biochemical activities other than antioxidant activity, or the possibility of such differences.

Response 5: Thanks for your kind suggestion. In the revised manuscript, we added ascorbic acid (Vc) as a positive control (in Figure 6), and some sentences for discussion were also replenished (in Lines 240-248).

Round 2

Reviewer 1 Report

non